# A protocol for the VISION study: An indiVidual patient data meta-analysis of randomised trials comparing MRI-targeted biopsy to standard transrectal ultraSound guided bIopsy in the detection of prOstate cancer

Veeru Kasivisvanathan[1,2]*, Vinson Wai-Shun Chan[1,3], Keiran D. Clement[4], Brooke Levis[5], Masoom Haider[6], Ridhi Agarwal[7], Mark Emberton[1,2,8], Gregory R. Pond[9], Yemisi Takwoingi[7], Laurence Klotz[10‡], Caroline M. Moore[1,2‡], VISION study collaborators¶

1 Division of Surgery and Interventional Sciences, University College London, United Kingdom, 2 Department of Urology, University College London Hospitals Trust, London, United Kingdom, 3 School of Medicine, Faculty of Medicine and Health, University of Leeds, Leeds, United Kingdom, 4 Department of Urology, NHS Greater Glasgow & Clyde, Glasgow, United Kingdom, 5 Centre for Prognosis Research, School of Medicine, Keele University, Staffordshire, United Kingdom, 6 Toronto General Hospital, Department of Radiology, University of Toronto, Toronto, Ontario, Canada, 7 Test Evaluation Research Group, Institute of Applied Health Research, University of Birmingham, and NIHR Birmingham Biomedical Research Centre, University Hospitals Birmingham NHS Foundation Trust and University of Birmingham, Birmingham, United Kingdom, 8 NIHR UCLH/UCL Comprehensive Biomedical Research Centre, London, United Kingdom, 9 Department of Biostatistics, McMaster University, Hamilton, Ontario, Canada, 10 Division of Urology, Sunnybrook Health Sciences Centre, University of Toronto, Toronto, Ontario, Canada

‡ LK and CMM share joint senior authorship on this work.
¶ Membership of the VISION study collaborators is listed in the Acknowledgments.
* veeru.kasi@ucl.ac.uk

**Funding:** The author(s) received no specific funding for this work, however the authors would

## Abstract

### Background

Transrectal ultrasound (TRUS) guided biopsy for prostate cancer is prone to random and systemic error and has been shown to have a negative predictive value of 70%. PRECISION and PRECISE are among the first randomised studies to evaluate the new MRI-targeted biopsy (MRI-TB) pathway with a non-paired design to detect clinically significant prostate cancer and avoid unnecessary treatment. The trials' results individually demonstrated non-inferiority of MRI-TB compared to TRUS biopsy. An individual patient data (IPD) meta-analysis was planned from the outset of the two trials in parallel and this IPD meta-analysis aims to further elucidate the utility of MRI-TB as the optimal diagnostic pathway for prostate cancer.

### Methods and materials

This study is registered on PROSPERO (CRD42021249263). A search of Medline, Embase, Cochrane Central Register of Registered Trials (CENTRAL), Web of Science, and ClinicalTrials.gov was performed up until 4th February 2021. Only randomised controlled trials (PRECISE, PRECISION and other eligible trials) comparing the MRI-targeted biopsy

like to acknowledge the following: Veeru Kasivisvanathan is an Academic Clinical Lecturer funded by the United Kingdom National Institute for Health Research (NIHR). Brooke Levis is funded by a Fonds de Recherche du Québec - Santé Postdoctoral Training Fellowship. Yemisi Takwoingi is funded by a UK NIHR Postdoctoral Fellowship and supported by the NIHR Birmingham Biomedical Research Centre. The views expressed in this publication are those of the author(s) and not necessarily those of the NHS, the NIHR or the Department of Health and Social Care.

**Competing interests:** I have read the journal's policy and the authors of this manuscript have the following competing interests: Veeru Kasivisvanathan is an Academic Clinical Lecturer funded by the United Kingdom National Institute for Health Research (NIHR). Brooke Levis is funded by a Fonds de Recherche du Québec - Santé Postdoctoral Training Fellowship. Yemisi Takwoingi is funded by a UK NIHR Postdoctoral Fellowship and supported by the NIHR Birmingham Biomedical Research Centre. The views expressed in this publication are those of the author(s) and not necessarily those of the NHS, the NIHR or the Department of Health and Social Care. This does not alter our adherence to PLOS ONE policies on sharing data and materials.

pathway and traditional TRUS biopsy pathway will be included. The primary outcome of the review is the proportion of men diagnosed with clinically significant prostate cancer in each arm (Gleason $\geq$ 3+4 = 7). IPD and study-level data and characteristics will be sought from eligible studies. Analyses will be done primarily using an intention-to-treat approach, and a one-step IPD meta-analysis will be performed using generalised linear mixed models. A non-inferiority margin of 5 percentage points will be used. Heterogeneity will be quantified using the variance parameters from the mixed model. If there is sufficient data, we will investigate heterogeneity by exploring the effect of the different conducts of MRIs, learning curves of MRI reporting and MRI targeted biopsies.

## Trial registration

This systematic review is registered on PROSPERO (CRD42021249263)

---

# 1. Introduction

## 1.1. Rationale and objectives

In men with a suspicion of localised prostate cancer (e.g. raised PSA and/or abnormal digital rectal exam and/or family history of prostate cancer), the traditional standard of care for diagnosis is systematic transrectal ultrasound guided (TRUS) prostate biopsy. During TRUS biopsy, 10–12 cores are taken randomly. Ultrasound traditionally does not visualize prostate cancer well, therefore TRUS biopsy is prone to random and systematic error and has been shown to have a negative predictive value of just 74% for detection of lesions with Gleason $\geq$ 4 +3 and/ or cancer core length $\geq$ 6mm [1, 2]. Although negative predictive value is dependent on inclusion criterion for consideration of biopsy and other factors such as what one accepts to be deemed a clinically significant cancer, this could lead to the possibility of incorrect risk stratification of patients, leading to poorly informed treatment decisions.

An alternative method of diagnosing cancer in men with suspicion of prostate cancer is to perform a multi-parametric MRI (mpMRI) scan of the prostate [3]. mpMRI can identify suspicious lesions in the prostate with a reported negative predictive value of 85–95% for the identification of clinically significant cancer [4, 5], though it does require expertise in the conduct and reporting of images [6]. If an MRI is suspicious (typically with a score 3–5 on a 5-point Likert scale from the PIRADs v2 score [7]), a biopsy procedure must still currently be carried out to provide confirmatory pathology, and, if cancer is present, to confirm Gleason Grade to guide treatment-based decisions and prognostic discussions [6].

A recent systematic review has demonstrated that MRI-targeted biopsy (MRI-TB) alone can detect more clinically significant cancer and less clinically insignificant cancer than systematic biopsy with a higher sampling efficiency [8]. However, the majority of the evidence from previous systematic reviews is from studies comparing patients who underwent both biopsy techniques (MRI-targeted biopsy and TRUS-biopsy) in the same sitting [8]. This within-patient design is subject to incorporation bias due to the potential knowledge of location of MRI lesions during TRUS biopsy [8], however, 9 out of 68 studies (13%) included in the systematic review was performed by the same operator, suggesting no blinding in 87% of the studies [8]. Furthermore, this within-patient design also limits the comparison of patients who have an MRI lesion (and have both MRI-targeted biopsy and standard TRUS biopsy) which does not capture outcomes in patients who have non-suspicious MRIs (Likert score 1 or 2), who may not routinely be offered biopsy [8].

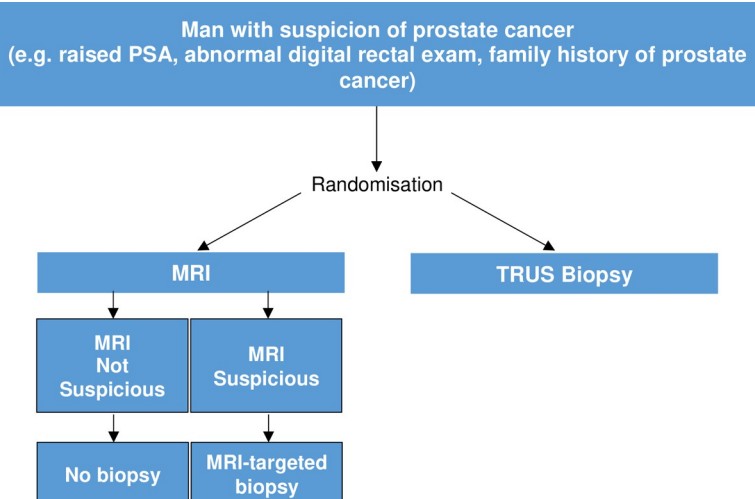

**Fig 1. Study design of PRECISION, PRECISE and other potentially included studies in this review.**

Therefore, to establish the diagnostic yield of MRI-targeted biopsy compared to standard TRUS biopsy alone, the PRECISION trial [3], an international study led by University College London in the United Kingdom and the PRECISE trial [9], a national study led by Sunnybrook Health Sciences Centre in Canada, were designed in parallel from the outset using a similar protocol [10] deliberately with an end goal of performing an individual patient data (IPD) meta-analysis to avoid the above biases. Both trials randomised patients with suspicion of prostate cancer who are "biopsy-naïve" to either (a) standard TRUS biopsy or (b) MRI with or without targeted biopsy (MRI-targeted biopsy). In the standard TRUS biopsy arm, men underwent standard TRUS biopsy alone. In the MRI-targeted biopsy arm, men with a non-suspicious MRI result avoided biopsy, whereas men who had lesions scoring 3,4 or 5 on the PIRADS v2 score [7] underwent MRI-targeted biopsy alone (without the addition of standard TRUS biopsy cores) (Fig 1). Whilst the PRECISE study showed that MRI with or without targeted biopsy in men with MRI-targetable lesions was non-inferior to TRUS biopsy for the detection of clinically significant cancer, the PRECISION study demonstrated that the MRI pathway was superior. Meta-analysis may help to establish whether the MRI-pathway is indeed non-inferior or superior to the TRUS biopsy pathway. In addition, further clarification on pathological and clinical outcomes that the individual studies were not powered to evaluate may be possible.

The VISION study aims to consolidate evidence for the optimal way to investigate biopsy naïve men with suspected prostate cancer. The results may influence international policies and guidelines' recommendations.

## 2. Material and methods

This protocol was prepared according to the Preferred reporting items for systematic review and meta-analysis protocols (PRISMA-P) 2015 statement (S1 Appendix) [11]. The study will be performed following the recommendations of the PRISMA guidelines [12] and relevant components of the IPD meta-analysis [13] and diagnostic test accuracy [14] extensions. As the intended studies for inclusion are studies of diagnostic yield rather than diagnostic accuracy, it is anticipated that some of the items in PRISMA-DTA will not be applicable. The three relevant checklists are presented in the S1 Appendix.

## 2.1. Review question

In biopsy-naïve men with clinical suspicion of prostate cancer, what proportion are diagnosed with clinically significant cancer by a strategy of MRI with targeted biopsy (TB) in suspicious MRIs and no biopsy in non-suspicious MRIs (MRI±TB) compared to a strategy of standard TRUS biopsy?

**2.1.1 Population.** Only studies reporting biopsy-naïve men with clinical suspicion of localized prostate cancer (i.e raised PSA and/or abnormal DRE and/or family history of prostate cancer) and advised to have a prostate biopsy will be included. Studies reporting on patients who have undergone previous prostate biopsy or prior treatment for prostate cancer will be excluded. Patients under the age of 18, patients with contraindications to MRI and patients with contraindications to prostate biopsy shall also be excluded.

**2.1.2. Intervention.** Studies with patients undergoing MRI will be included if those with suspicious MRI underwent targeted biopsy alone, and those with non-suspicious lesion avoided biopsy. Studies investigating either transrectal and transperineal approach for targeted biopsies will be included.

**2.1.3. Comparator.** Studies reporting patients receiving only standard TRUS biopsy as a comparator will be included.

**2.1.4. Outcomes.** Primary outcomes:

- The proportion of men diagnosed with clinically significant cancer (Gleason 3+4 or Gleason Grade Group 2 or greater)

    Secondary outcomes:

1. Proportion of men diagnosed with clinically insignificant cancer (Gleason 6 or less or Gleason Grade Group 1)

2. Proportion of men who avoided prostate biopsy following MRI

3. Proportion of men with clinically significant cancer, clinically insignificant cancer and no cancer by PIRADs v2 score

4. Proportion of biopsy cores positive for cancer for MRI-TB compared to systematic TRUS biopsy

5. Cancer core length of the most involved biopsy core

6. Proportion of men who go onto definitive local or systemic treatment for prostate cancer

7. Proportion of Gleason Grade upgrading in men undergoing radical prostatectomy

8. Proportion of men with post-biopsy adverse events

9. Health-related quality of life scores

10. Predictive factors for clinically significant cancer detection

11. Proportion of men diagnosed with Gleason grade group 3 cancer or higher

## 2.2. Study selection criteria

**2.2.1. Inclusion criteria.** Included studies must be randomized controlled trials that satisfy the population, intervention, comparator, and outcomes as reported above.

**2.2.2. Exclusion criteria.** Studies with a paired cohort design will be excluded. Conference proceedings or abstracts will also be excluded if no eventual full publication is available in the literature or no aggregate data or IPD is available after contacting the corresponding author.

## 2.3. Information sources and search strategy

This is a review based on a previous systematic review and meta-analysis [8] with searches performed up until the 28th July 2017. In the previous systematic review, only one study eligible for the present review was identified [3]. Hence, the searches for this review were performed from 28th July 2017 to 4th February 2021 on MEDLINE, EMBASE, Web of Science, Cochrane Central of Registered Trials (CENTRAL) and ClinicalTrials.gov using a combination of Medical Subject Heading (MeSH) Terms and keywords. No language restrictions and other limits were imposed on the study. The full search strategy is presented in the S1 Appendix.

## 2.4. Screening and data extraction

**2.4.1. Data management.**    After the systematic literature search is carried out, all references will be uploaded into Endnote, a reference manager software [15]. De-duplication will be performed. The remaining articles will be imported into Covidence [16] for abstract, title and full-text screening.

**2.4.2. Selection process.**    Every title and abstract will be assessed independently by two reviewers (V.W.S.C, K.D.C). In the event of disagreement, consensus will be attempted after discussion between the pair of screeners. If consensus cannot be reached, a third author (V.K) will resolve any differences. After title and abstract screening, full-texts for potentially eligible studies will be retrieved for further assessment in a similar manner. Reviewers will not be blinded to study authors, institution, publication journal or year of publication.

**2.4.3. Data collection process.**    Where studies are identified from full text review as meeting the inclusion and exclusion criteria, study-level and individual patient-level data will be sought. For study-level data, this will be extracted by two independent authors. Where the data is not reported, the study's corresponding author will be contacts for further information. For patient-level data, authors of each study will be contacted to provide the original patient-level data. For studies deemed to meet our inclusion criterion that we are unable to obtain IPD for, we will consider undertaking a secondary aggregate analysis to combine the IPD meta-analysis results with published results from studies that did not contribute to the IPD meta-analysis.

Data extracted will include characteristics of the studies, baseline patient characteristics (e.g. PSA level), the arm of investigation and the outcomes of the patients. The full codebook planned is presented in the S1 Appendix.

## 2.5. Assessment of risk of bias and reporting standards in individual studies

Individual studies will be assessed for risk of bias and applicability by two independent authors using the QUADAS-2 and QUADAS-C tool domains [17, 18] (S1 Appendix) where relevant. In addition, we will augment the risk of bias assessment by using the Cochrane risk of bias 2.0 tool for RCTs [19] (S1 Appendix). Individual studies will also be assessed according to their reporting standards using the START Criteria for MRI-targeted biopsy studies [20] (S1 Appendix).

## 2.6. Data synthesis

For each study, study-level and participant-level characteristics will be presented for the MRI arm and TRUS-biopsy arm to explore how patients differ between trials. This will also enable comparison of the characteristics of included studies with those for which we are unable to obtain IPD. For studies where IPD is obtained, continuous variables will be described using the mean and standard deviation, or median and interquartile range as appropriate.

Categorical variables will be described using frequencies and percentages. If required, variables within IPD datasets will be standardised to ensure common scales/measurements across the included studies. Original data will also be reanalysed to compare with published results to validate the IPD datasets.

**2. 6. 1. Analysis of primary outcome.** All randomised patients with outcome data and studies with available IPD will be included in the analysis. We will use an intention-to-treat approach to compare the proportion of men diagnosed with clinically significant cancer (Gleason 3+4 or Gleason Grade Group 2 or greater) in the MRI arm and TRUS-biopsy arm. A one-step IPD meta-analysis will be performed using a generalised linear mixed model (GLMM) to analyse all studies simultaneously while accounting for the clustering of participants within each study (random intercept model). We will use an identity link function in the model to allow estimation of the absolute difference in the proportion of men diagnosed with clinically significant cancer between the two arms. Similar to the PRECISION and PRECISE trials, a non-inferiority margin of 5 percentage points will be used to assess non-inferiority. This margin was determined at an expert consensus group meeting [20]. If the lower limit of the two-sided 95% confidence interval for the difference in the proportion of clinically significant cancer in the MRI arm relative to the TRUS-biopsy arm is greater than –5 percentage points, then MRI will be considered non-inferior to TRUS biopsy alone. In addition, if the lower limit exceeds zero, superiority will be inferred.

**2. 6. 2. Analysis of secondary outcomes.** The following secondary outcomes comparing the proportion in the MRI arm and TRUS-biopsy arm will be analysed using a one-step IPD meta-analysis, as above:

i. Proportion of men diagnosed with clinically insignificant cancer (Gleason 6 or less or Gleason grade group 1)

ii. Proportion of men who go onto definitive treatment for prostate cancer

iii. Proportion of men undergoing radical prostatectomy with Gleason grade upgrading

iv. Proportion of men diagnosed with Gleason Grade group 3 or higher

v. Proportion of biopsy cores positive for cancer for MRI-TB compared to systematic TRUS biopsy

For the ordinal secondary outcome comparing the proportion of men in the MRI arm and TRUS-biopsy arm diagnosed with clinically significant cancer, clinical insignificant cancer and no cancer by PIRADsv2 score, a one-step IPD meta-analysis using a GLMM with an identity link function with ordinal distribution will be applied.

For the continuous secondary outcomes comparing the cancer core length of the most involved biopsy core and the health-related quality of life in the MRI arm and TRUS-biopsy arm, mean differences will be meta-analysed using a linear mixed model with a random intercept. It is unlikely that data on adverse events will be adequate for meta-analysis and so we will present these descriptively using frequencies and percentages.

Heterogeneity across studies will be evaluated using the variance parameters from the random-effects models. Data permitting, subgroup analyses will be performed for the primary outcome for each potential source of heterogeneity by extending the relevant GLMM to assess the interaction with trial arm. Potential sources of heterogeneity include the conduct and reporting of mpMRI and systematic or MRI-targeted biopsy procedures as types of MRI machines and learning curves of MRI reporters and biopsy-operators may affect the outcome of MRI targeted biopsies.

Depending on data availability, we plan to explore the following sources of heterogeneity:

i. mpMRI conduct

 a. The ability of MRI machines may vary amongst centres in identifying suspicious areas in the prostate dependent on sequences used and quality of the machine.

ii. Multiparametric MRI reporting

 a. There is a steep learning curve for interpreting MRI of the prostate by the radiologist, and the different expertise and experience may affect the ability of the radiologist to identify suspicious areas of the prostate.

iii. MRI-Targeted biopsy procedure

 a. Operators may have varying experience in carrying out the procedure and therefore different ability to target suspicious areas of the prostate.

 b. Methods of registration such as visual or software registration may have different ability to target suspicious areas of the prostate.

 c. Access route (transperineal or transrectal) may have different performance characteristics.

Sensitivity analyses will be performed for the primary outcome using a modified intention-to-treat and per-protocol approaches as outlined in the PRECISION study (S1 Appendix). For studies that meet our inclusion criterion but are unable to provide IPD, we will undertake a secondary analysis to combine the IPD meta-analysis results with published aggregate results, using a two-step IPD meta-analysis. The rationale for this analysis is to assess whether the meta-analysis results of the primary outcome is representative of those based on all the data, thus investigating data availability bias.

## 2.7. Meta-biases and confidence in cumulative evidence

Meta-biases such as publication bias will be investigated for asymmetry using a funnel plot if ten or more studies are available. The random-effects model will also be used to account for any variability between studies. The GRADE approach will not be used to evaluate the confidence of evidence given the nature of this IPD meta-analysis project and the question being investigated.

## Supporting information

**S1 Appendix. Supplementary appendix and PRISMA-P checklist.**
(DOCX)

## Acknowledgments

VISION study group collaborators
 Antti S. Rannikko, Helsinki University and Helsinki University Hospital, Helsinki
 Marcelo Borghi, Centro de Urología, Buenos Aires
 Valeria Panebianco, Sapienza University, Rome, Italy
 Lance A. Mynderse, Mayo Clinic, Rochester, MN, USA
 Markku H. Vaarala, Medical Research Center Oulu, University of Oulu and Oulu University Hospital, Oulu, Finland
 Alberto Briganti, Vita-Salute San Raffaele University, Milan, Italy
 Lars Budäus, Martini Klinik, Hamburg, Italy

Giles Hellawell, London North West Healthcare NHS Trust, London, United Kingdom

Richard G. Hindley, Hampshire Hospitals NHS Foundation Trust, Basingstoke, United Kindgom

Monique J. Roobol, Erasmus University Medical Center, Rotterdam, Netherlands

Scott Eggener, University of Chicago, Chicago, USA

Maneesh Ghei, Whittington Health NHS Trust, London, United Kingdom

Arnauld Villers, Université de Lille and Centre Hospitalier Universitaire Lille, Lille, France

Franck Bladou, Jewish General Hospital, Montreal, Canada

Geert M. Villeirs, Ghent University Hospital, Ghent, Belgium

Jaspal Virdi, Princess Alexandra Hospital NHS Trust, Harlow, United Kingdom

Silvan Boxler, University Hospital Bern, Bern, Switzerland

Grégoire Robert, Université de Bordeaux and Bordeaux Pellegrin University Hospital, Bordeaux, France

Paras B. Singh, Royal Free London NHS Foundation Trust, London, United Kingdom

Wulphert Venderink, Radboud University Medical Center, Nijmegen, Netherlands

Boris A Hadaschik, University Hospital Essen, Essen, and University Hospital Heidelberg, Heidelberg, Germany

Alain Ruffion, Hospices Civils de Lyon, Centre Hospitalier Lyon-Sud, France

Jim C. Hu, Weill Cornell Medicine, New York–Presbyterian Hospital, New York, USA

Daniel Margolis, Weill Cornell Medicine, New York–Presbyterian Hospital, New York, USA

Sébastien Crouzet, Hospices Civils de Lyon of the Hôpital Edouard Herriot, France

Samir S. Taneja, New York University Langone Medical Center, New York, USA

Peter Pinto, National Institutes of Health, Bethesda, MD, USA

Inderbir Gill, University of Southern California Institute of Urology, Keck School of Medicine, Los Angeles, USA

Clare Allen, University College London (UCL) and UCL Hospitals NHS Foundation Trust, London, United Kingdom

Francesco Giganti, University College London (UCL) and UCL Hospitals NHS Foundation Trust, London, United Kingdom

Alex Freeman, University College London (UCL) and UCL Hospitals NHS Foundation Trust, London, United Kingdom

Shonit Punwani, University College London (UCL) and UCL Hospitals NHS Foundation Trust, London, United Kingdom

Norman R. Williams, UCL Surgical and Interventional Trials Unit, London, United Kingdom

Chris Brew-Graves, UCL Surgical and Interventional Trials Unit, London, United Kingdom

Joseph Chin, London Health Sciences Centre, University of Western Ontario, London, Ontario, Canada

Peter Black, Vancouver Prostate Centre, Department of Urologic Sciences, The University of British Columbia, Vancouver, British Columbia, Canada

Antonio Finelli, Princess Margaret Hospital, University of Toronto, Toronto, Ontario, Canada

Maurice Anidjar, Jewish General Hospital, McGill University, Montreal, Québec, Canada

Frank Bladou, Division of Urology, Sunnybrook Health Sciences Centre, University of Toronto, Toronto, Ontario, Canada and Universite de Bordeaux, Bordeaux, France

Ashley Mercado, Vancouver Prostate Centre, Department of Urologic Sciences, The University of British Columbia, Vancouver, British Columbia, Canada

Mark Levental, Jewish General Hospital, McGill University, Montreal, Québec, Canada

Sangeet Ghai, Princess Margaret Hospital, University of Toronto, Toronto, Ontario, Canada

Sylvia D. Chang, Department of Radiology, University of British Columbia, Vancouver, British Columbia, Canada

Laurent Milot, Body and VIR Radiology Department, Hospices Civils de Lyon, Hospital Edouard Herriot, Lyon, France

Chirag Patel, Division of Urology, Sunnybrook Health Sciences Centre, University of Toronto, Toronto, Ontario, Canada

Zahra Kassam, London Health Sciences Centre, University of Western Ontario, London, Ontario, Canada

Andrew Loblaw, Institute of Healthcare Policy and Management, Department of Radiation Oncology, Ontario Institute of Cancer Research, University of Toronto, Toronto, Ontario, Canada

Marlene Kebabdjian, Division of Urology, Sunnybrook Health Sciences Centre, University of Toronto, Toronto, Ontario, Canada

Craig C. Earle, Ontario Institute of Cancer Research, Toronto, Ontario, Canada

Alexander Ng, UCL Medical School, London, UK

Aqua Asif, Leicester Medical School, Leicester, UK and Division of Surgery and Interventional Sciences, University College London, UK

## Author Contributions

**Conceptualization:** Veeru Kasivisvanathan.

**Methodology:** Veeru Kasivisvanathan, Vinson Wai-Shun Chan, Keiran D. Clement, Brooke Levis, Masoom Haider, Ridhi Agarwal, Mark Emberton, Gregory R. Pond, Yemisi Takwoingi, Laurence Klotz, Caroline M. Moore.

**Project administration:** Vinson Wai-Shun Chan.

**Supervision:** Veeru Kasivisvanathan, Yemisi Takwoingi, Laurence Klotz, Caroline M. Moore.

**Validation:** Veeru Kasivisvanathan.

**Writing – original draft:** Veeru Kasivisvanathan, Vinson Wai-Shun Chan, Keiran D. Clement.

**Writing – review & editing:** Veeru Kasivisvanathan, Vinson Wai-Shun Chan, Keiran D. Clement, Brooke Levis, Masoom Haider, Ridhi Agarwal, Mark Emberton, Gregory R. Pond, Yemisi Takwoingi, Laurence Klotz, Caroline M. Moore.

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
