## [Decision Letter · Decision Letter 0]

7 Dec 2021

PONE-D-21-31061A protocol for the VISION study: An indiVidual patient data meta-analysis of randomised trials comparing MRI-targeted biopsy to standard transrectal ultraSound guided bIopsy in the detection of prOstate cancerPLOS ONE

Dear Dr. Kasivisvanathan,

Thank you for submitting your manuscript to PLOS ONE. After careful consideration, we feel that it has merit but does not fully meet PLOS ONE’s publication criteria as it currently stands. Therefore, we invite you to submit a revised version of the manuscript that addresses the points raised during the review process.

We look forward to receiving your revised manuscript.

Kind regards,

Kim Moretti

Academic Editor

PLOS ONE

Journal Requirements:

“I have read the journal's policy and the authors of this manuscript have the following competing interests: Veeru Kasivisvanathan is an Academic Clinical Lecturer funded by the United Kingdom National Institute for Health Research (NIHR). Brooke Levis is funded by a Fonds de Recherche du Québec - Santé Postdoctoral Training Fellowship. Yemisi Takwoingi is funded by a UK NIHR Postdoctoral Fellowship and supported by the NIHR Birmingham Biomedical Research Centre. The views expressed in this publication are those of the author(s) and not necessarily those of the NHS, the NIHR or the Department of Health and Social Care.”

4. One of the noted authors is a group or consortium [VISION study collaborators]. In addition to naming the author group, please list the individual authors and affiliations within this group in the acknowledgments section of your manuscript. Please also indicate clearly a lead author for this group along with a contact email address.

5. We note that this manuscript is a systematic review or meta-analysis; our author guidelines therefore require that you use PRISMA guidance to help improve reporting quality of this type of study. Please upload copies of the completed PRISMA checklist as Supporting Information with a file name “PRISMA checklist

Reviewers' comments:

Reviewer's Responses to Questions

**Comments to the Author**

1. Does the manuscript provide a valid rationale for the proposed study, with clearly identified and justified research questions?

Reviewer #1: Yes

Reviewer #2: Partly

Reviewer #3: Yes

2. Is the protocol technically sound and planned in a manner that will lead to a meaningful outcome and allow testing the stated hypotheses?

Reviewer #1: Yes

Reviewer #2: Yes

Reviewer #3: Yes

3. Is the methodology feasible and described in sufficient detail to allow the work to be replicable?

Reviewer #1: Yes

Reviewer #2: Yes

Reviewer #3: Yes

4. Have the authors described where all data underlying the findings will be made available when the study is complete?

Reviewer #1: No

Reviewer #2: Yes

Reviewer #3: Yes

5. Is the manuscript presented in an intelligible fashion and written in standard English?

Reviewer #1: Yes

Reviewer #2: Yes

Reviewer #3: Yes

6. Review Comments to the Author

You may also provide optional suggestions and comments to authors that they might find helpful in planning their study.

Reviewer #1: In this study authors present their plan to conduct an individual patient data (IPD) meta-analysis to elaborate the utility of MRI-targeted biopsy as an optimal diagnostic pathway for prostate cancer. The study plan is solid and well-written. I have a few comments that I the authors should address to qualify for following up with the journal for further review of their findings.

- The authors have discussed limitations of a previously published systematic-review on this topic. One noted point is that a within-patient design of that study is subject to bias due to the potential knowledge of location of MRI lesions during TRUS biopsy. However, data in the afore-mentioned review is mainly obtained from studies with blinded design. Using within-patient design may be superior as it can address confounders in prostate cancer diagnosis. Authors should further justify their rational for superiority of of their meta-analysis compared to the within-patient studies with blinded design.

. This within-patient design is subject to incorporation

- Beside the diagnostic accuracy of each modality, the overall rate of prostate cancer depend on several factors, such as age and PSA level etc. - How will the authors address the heterogeneous baseline characteristics of the individual studies included here?

- The authors have mentioned that authors of the included studied for their meta-analysis will be contacted to provide patient-level data. How would not receiving a response be addressed in this study.

Reviewer #2: Thank you for the opportunity to review: A protocol for the VISION study: An indiVidual patient data meta-analysis of randomised trials comparing MRI-targeted biopsy to standard transrectal ultraSound guided bIopsy in the detection of prOstate cancer

I would consider changing the acronym because we just had another VISION Trial published in prostate cancer, but that’s just me.

The overall impression of the study protocol is good, but I have some minor concerns.

Abstract

I find the abstract somewhat vague. Is the intent of the analysis to analyse PRECISION and PRECISE and/or to include other RCTs?

I don’t agree with the 70% NPV statement of systematic TRUS biopsy, which is also stated in the introduction. The NPV depends heavily on the inclusion criteria and what you define as NPV. A true negative patient or a patient with GS<7? Death from PCa? Systematic TRUS biopsy have an NPV of 0 in men with high age, PSA>100 and clinical T3/4 tumors. There is a tendency to underplay systematic TRUS biopsy as a diagnostic procedure which is not true. It may miss cancers in certain clinical scenarios, but that is not necessarily bad, just look at https://pubmed.ncbi.nlm.nih.gov/31411967/

Introduction

I think reference 1 is bad for a statement on the NPV of TRUS biopsy. That reference is personal comment. Also, what do you mean by NPV here? That no cancer at all is there? Again, I think it is important to acknowledge that systematic TRUS biopsy has high performance for diagnosing both the cancer, GG2 cancers and even more advanced cancers when you combine age, PSA and clinical T-category.

In the next paragraph, NPV now refers to the presence of non-significant PCa.

“Whilst the PRECISE study showed that MRI with or without targeted biopsy was non-inferior to TRUS biopsy,”

I am not sure I understand that sentence. The trial compared target biopsies for MRI PIRADs 3-5 vs systematic TRUS biopsy.

“Meta-analysis may help to establish whether the MRI-pathway is indeed non inferior or superior to the TRUS biopsy pathway. In addition, further clarification on pathological and clinical outcomes that the individual studies were not powered to evaluate may be possible”

I am not sure whether I understand from the introduction that this is a meta-analysis of two trials or the two trials + other trials. If you believe that you can argue superiority of MRI+ target biopsy to systematic TRUS biopsy from PRECISION that was a non-inferiority trial that concluded statistical superiority in the post-hoc analysis and a trial that was only non-inferior I think the reader needs a little more background on how that is possible.

Methods

“…and no biopsy in non-suspicious MRIs (MRI±TB) compared to a strategy of standard TRUS biopsy?”

I find that question a little bit self-conclusive. The problem with the trials that you are going to look into is that you are up-front comparing apples and pears. The two trials never took biopsies in patients with PIRADs 1 and 2 (at least). I don’t think its fair to compare biopsy vs no-biopsy.

Problematically, the PRECISION trial included men that some would never biopsy at all, even order an MRI scan. Most patients had cT1, no family history of PCa, median age of 65 and PSA og 6,7. So its rather easy to say that if you take biopsies you find something that you wouldn’t have liked to find in that kind of man, just given the prevalence of the disease in the age group.

Inclusion criteria

As mentioned previously, I don’t understand why you just don’t state that you intend to include one more trial and update a previous publication. Because we all know that there is only one more trial within the narrow research question of MRI + target vs systematic TRUS biopsy

Reviewer #3: The proposed study protocol seems very sensible and sets out to help answer and important set of questions. The team are very experienced with this topic and will I believe deliver a great study.

7. PLOS authors have the option to publish the peer review history of their article (what does this mean?). If published, this will include your full peer review and any attached files.

Reviewer #1: No

Reviewer #2: **Yes: **M. Andreas Røder

Reviewer #3: **Yes: **Tim Dudderidge

---

## [Author Response · Author response to Decision Letter 0]

13 Jan 2022

Reviewer #1: In this study authors present their plan to conduct an individual patient data (IPD) meta-analysis to elaborate the utility of MRI-targeted biopsy as an optimal diagnostic pathway for prostate cancer. The study plan is solid and well-written. I have a few comments that I the authors should address to qualify for following up with the journal for further review of their findings.

- The authors have discussed limitations of a previously published systematic-review on this topic. One noted point is that a within-patient design of that study is subject to bias due to the potential knowledge of location of MRI lesions during TRUS biopsy. However, data in the afore-mentioned review is mainly obtained from studies with blinded design. Using within-patient design may be superior as it can address confounders in prostate cancer diagnosis. Authors should further justify their rational for superiority of of their meta-analysis compared to the within-patient studies with blinded design.

Authors’ response: Thank you very much for your comments. There are pros and cons for a within-patient design, as well as a randomised design. Whilst one design won’t necessarily be superior to the other, it is fair to say both are considered level 1 evidence in this setting. The systematic review by Kasivisvanathan et al demonstrated that 9 of the 68 (13%) studies of the within patient design had a different operator performing MR-targeted and systematic core biopsies. This means there was no blinding in 87% of studies. Due to incorporation bias, the performance of MRI targeted biopsy can influence the performance of TRUS biopsy and vice versa. An advantage of the randomised study design is that performance of one test will not be influenced by the other. This is demonstrated by the PRECISION study which showed a high detection ratio in favour of targeted alone biopsy. This was the only study included in this systematic review which performed targeted biopsy alone in the MRI arm. 

- Beside the diagnostic accuracy of each modality, the overall rate of prostate cancer depend on several factors, such as age and PSA level etc. - How will the authors address the heterogeneous baseline characteristics of the individual studies included here?

Authors’ response: Thank you very much for the comments. It is possible that there may be only two included studies in the IPD meta-analysis (IPDMA) (PRECISE AND PRECISION). These two studies were designed from the outset with similar protocols to allow an IPDMA and as such have similar inclusion criterion (clinical suspicion of prostate cancer, PSA 20ng per millilitre or less, and no contraindications to MRI or biopsy). This protocol similarity should negate some of this heterogeneity. Furthermore, we will evaluate statistical heterogeneity across studies using the variance parameters from the random-effects model. Data permitting, subgroup analyses will be performed for the primary outcome for each potential source of heterogeneity by extending the relevant generalised linear mixed model (GLMM) to assess the interaction with trial arm as mentioned under 2.6. Data analysis – analysis of primary outcome. If there are any other studies identified, they will be examined for clinical heterogeneity, and if they are too great, they will not be meta-analysed.

- The authors have mentioned that authors of the included studied for their meta-analysis will be contacted to provide patient-level data. How would not receiving a response be addressed in this study.

Authors’ response: Thank you very much for the comments. For studies we do not receive a response for, we will consider undertaking a secondary aggregate analysis to combine the IPD meta-analysis results with published results from studies that did not contribute to the IPD meta-analysis. This is outlined under Protocol Section 2.4.3. 

Reviewer #2: Thank you for the opportunity to review: A protocol for the VISION study: An indiVidual patient data meta-analysis of randomised trials comparing MRI-targeted biopsy to standard transrectal ultraSound guided bIopsy in the detection of prOstate cancer

I would consider changing the acronym because we just had another VISION Trial published in prostate cancer, but that’s just me.

The overall impression of the study protocol is good, but I have some minor concerns.

Authors’ Response: Thank you very much for the comment. We have already published the PROSPERO registration with the name VISION and we hope to maintain consistency and keep VISION as our study acronym. Although the acronym is the same, the full study name is different. We hope that is ok for the reviewer. 

Abstract

I find the abstract somewhat vague. Is the intent of the analysis to analyse PRECISION and PRECISE and/or to include other RCTs?

Authors’ Response: Thank you for your comments. We intend to analyse any eligible RCTs with the same design and have altered the abstract to reflect this. 

I don’t agree with the 70% NPV statement of systematic TRUS biopsy, which is also stated in the introduction. The NPV depends heavily on the inclusion criteria and what you define as NPV. A true negative patient or a patient with GS<7? Death from PCa? Systematic TRUS biopsy have an NPV of 0 in men with high age, PSA>100 and clinical T3/4 tumors. There is a tendency to underplay systematic TRUS biopsy as a diagnostic procedure which is not true. It may miss cancers in certain clinical scenarios, but that is not necessarily bad, just look at https://pubmed.ncbi.nlm.nih.gov/31411967/

Introduction

I think reference 1 is bad for a statement on the NPV of TRUS biopsy. That reference is personal comment. Also, what do you mean by NPV here? That no cancer at all is there? Again, I think it is important to acknowledge that systematic TRUS biopsy has high performance for diagnosing both the cancer, GG2 cancers and even more advanced cancers when you combine age, PSA and clinical T-category.

In the next paragraph, NPV now refers to the presence of non-significant PCa.

Authors’ Response: Thank you, these are both fair comments. We have now adjusted the manuscript and the references to include the PROMIS study. According to the PROMIS study, the NPV for detecting clinically significant cancer (Gleason ≥ 4+3 or cancer core length ≥ 6mm) is 74%, and this number reduces to 65% and 63% when considering "Gleason ≥ 4+3 or cancer core length ≥ 4 “and “Any Gleason score 7 (≥3+4)”, respectively. The edited text now reads: “Ultrasound traditionally does not visualize prostate cancer well, therefore TRUS biopsy is prone to random and systematic error and has been shown to have a negative predictive value of just 74% for lesions with Gleason ≥ 4+3 and/or cancer core length ≥ 6mm (1,2). Although negative predictive value is dependent on inclusion criterion for consideration of biopsy and other factors such as what one accepts to be deemed a clinically significant cancer, this could lead to the possibility of incorrect risk stratification of patients, leading to poorly informed treatment decisions.”. 

“Whilst the PRECISE study showed that MRI with or without targeted biopsy was non-inferior to TRUS biopsy,”

I am not sure I understand that sentence. The trial compared target biopsies for MRI PIRADs 3-5 vs systematic TRUS biopsy.

Authors’ Response: Thank you for highlighting this, we have edited the sentence to make this clearer: “Whilst the PRECISE study showed that MRI with or without targeted biopsy in men with MRI-targetable lesions was non-inferior to TRUS biopsy for the detection of clinically significant cancer, the PRECISION study demonstrated that the MRI pathway was superior.”

“Meta-analysis may help to establish whether the MRI-pathway is indeed non inferior or superior to the TRUS biopsy pathway. In addition, further clarification on pathological and clinical outcomes that the individual studies were not powered to evaluate may be possible”

I am not sure whether I understand from the introduction that this is a meta-analysis of two trials or the two trials + other trials. If you believe that you can argue superiority of MRI+ target biopsy to systematic TRUS biopsy from PRECISION that was a non-inferiority trial that concluded statistical superiority in the post-hoc analysis and a trial that was only non-inferior I think the reader needs a little more background on how that is possible.

Authors’ Response: Thank you very much for your comments, this is a good question. This was discussed by key members of the Cochrane Diagnostic Test Accuracy group in the design of this study, and this was the conclusion that they have made. This relates to statistical interpretation of the confidence intervals. We have discussed this in more detail for the reader in the manuscript under 2.6. Data synthesis – analysis of primary outcome: “. Similar to the PRECISION and PRECISE trials, a non-inferiority margin of 5 percentage points will be used to assess non-inferiority. This margin was determined at an expert consensus group meeting (20). If the lower limit of the two-sided 95% confidence interval for the difference in the proportion of clinically significant cancer in the MRI arm relative to the TRUS-biopsy arm is greater than −5 percentage points, then MRI will be considered non-inferior to TRUS biopsy alone. In addition, if the lower limit exceeds zero, superiority will be inferred.”

Methods

“…and no biopsy in non-suspicious MRIs (MRI±TB) compared to a strategy of standard TRUS biopsy?”

I find that question a little bit self-conclusive. The problem with the trials that you are going to look into is that you are up-front comparing apples and pears. The two trials never took biopsies in patients with PIRADs 1 and 2 (at least). I don’t think its fair to compare biopsy vs no-biopsy.

Problematically, the PRECISION trial included men that some would never biopsy at all, even order an MRI scan. Most patients had cT1, no family history of PCa, median age of 65 and PSA og 6,7. So its rather easy to say that if you take biopsies you find something that you wouldn’t have liked to find in that kind of man, just given the prevalence of the disease in the age group.

Authors’ Response: Thank you for the comment. In this study design, as a proportion of patients in the MRI arm do not get biopsy, this actually makes it harder for the MRI arm to show non-inferiority for detection of clinically significant cancer than the TRUS biopsy arm, in which everyone gets a biopsy. Just to clarify, the hypothesis is that the MRI arm is non-inferior to the TRUS biopsy arm, not the other way round. We acknowledge that for clinically insignificant cancer detection, that the reviewer’s comments hold true, i.e fewer men biopsy leading to fewer insignificant cancers detected, however, we would consider this to be an advantage for a diagnostic approach. 

Inclusion criteria

As mentioned previously, I don’t understand why you just don’t state that you intend to include one more trial and update a previous publication. Because we all know that there is only one more trial within the narrow research question of MRI + target vs systematic TRUS biopsy

Authors’ Response: Thank you for your comments. The only way to know whether there is more than one additional trial is to perform a systematic review, as per the PRISMA guidelines for IPD meta-analysis. 

Reviewer #3: The proposed study protocol seems very sensible and sets out to help answer and important set of questions. The team are very experienced with this topic and will I believe deliver a great study.

Authors’ Response: Thank you for your comments.

---

## [Editor Report · Decision Letter 1]

18 Jan 2022

A protocol for the VISION study: An indiVidual patient data meta-analysis of randomised trials comparing MRI-targeted biopsy to standard transrectal ultraSound guided bIopsy in the detection of prOstate cancer

PONE-D-21-31061R1

Dear Dr. Kasivisvanathan,

We’re pleased to inform you that your manuscript has been judged scientifically suitable for publication and will be formally accepted for publication once it meets all outstanding technical requirements.

Kind regards,

Kim Moretti

Academic Editor

PLOS ONE
---

## [Editor Report · Acceptance letter]

24 Jan 2022

PONE-D-21-31061R1 

A protocol for the VISION study: An indiVidual patient data meta-analysis of randomised trials comparing MRI-targeted biopsy to standard transrectal ultraSound guided bIopsy in the detection of prOstate cancer  

Dear Dr. Kasivisvanathan:

I'm pleased to inform you that your manuscript has been deemed suitable for publication in PLOS ONE. Congratulations! Your manuscript is now with our production department. 

Kind regards, 

on behalf of

Professor Kim Moretti 

Academic Editor

PLOS ONE